# Crystal Structure Analysis and Characterization of NADP-Dependent Glutamate Dehydrogenase with Alcohols Activity from *Geotrichum candidum*

Jing Zhu [1,2,†], Hai Hou [2,3,†], Kun Li [1], Xiaoguang Xu [2], Chunmei Jiang [2], Dongyan Shao [2], Junling Shi [2,*] and Dachuan Yin [2,*]

1. School of Food Science, Xinyang Agriculture and Forestry University, New 24 Street of Yangshan New District, Xinyang 464000, China; zhujingcy@163.com (J.Z.); likunkm@163.com (K.L.)
2. Key Laboratory for Space Bioscience and Biotechnology, School of Life Sciences, Northwestern Polytechnical University, 127 Youyi West Road, Xi'an 710072, China; houhai@nwpu.edu.cn (H.H.); xuxiaoguang@mail.nwpu.edu.cn (X.X.); jiangcm@nwpu.edu.cn (C.J.); shaoyan@nwpu.edu.cn (D.S.)
3. Institute of Medical Research, Northwestern Polytechnical University, 127 Youyi West Road, Xi'an 710072, China
* Correspondence: sjlshi2004@nwpu.edu.cn (J.S.); yindc@nwpu.edu.cn (D.Y.)
† These authors contributed equally to this work.

**Abstract:** To better understand its mechanism of activity towards higher alcohols, we overexpressed and purified new *Geotrichum candidum* GDH (*Gc*GDH). The purified *Gc*GDH (50.27 kDa) was then crystallized, and the crystal diffracted to a resolution of 2.3 Å using X-ray diffraction. We found that the *Gc*GDH crystal structure belonged to space group P212121 and was comprised of two hexamers organized into an asymmetric unit, with each subunit consisting of 452 amino acid residues. The binding sites between higher alcohols or L-glutamic acid and *Gc*GDH were consistent. The optimal reaction conditions for *Gc*GDH and hexanol were a pH of 4.0 and temperature of 30 °C, and those for *Gc*GDH and monosodium glutamate (MSG) were a pH of 8.0 and temperature of 20 °C. The $K$m values for hexanol and MSG were found to be 74.78 mM and 0.018 mM, respectively. Mutating *Gc*GDH Lys 113 to either Ala or Gly caused a dramatic reduction in its catalytic efficiency towards both MSG and hexanol, suggesting that Lys 113 is essential to the active site of *Gc*GDH.

**Keywords:** *Geotrichum candidum*; glutamate dehydrogenase; gene cloning; characterization; crystal structure

## 1. Introduction

Glutamate overstimulation may promote neurologic disorders such as Alzheimer's disease [1] and a set of adverse reactions known as MSG symptom complex, which includes headache, numbness, and palpitations [2]. Considering the widespread use of monosodium glutamate (MSG) in food products, there is increasing scientific interest in the use of glutamate dehydrogenase (GDH, EC 1.4.1.3) for monitoring glutamate contents in foods and biological samples. GDH contributes to the increase in the fruity flavor during ripening processes by inducing glutamate synthesis [3]. Downregulated GDH expression may reduce amino acid levels and remove the undesirable flavor of natto [4]. GDH has long been recognized as a key enzyme that affects ammonia production and plays an important role in nitrogen and carbon metabolism [5].

GDH is an enzyme that exists widely in the microbes and mitochondria of eukaryotes. It is responsible for the NAD(P)+/NAD(P)H-linked conversion of glutamate to α-ketoglutaric acid during carbon and nitrogen metabolism [4]. However, we previously found that *Gc*GDH from *Geotrichum candidum* S12 exhibits hexanol dehydrogenase activity to convert hexanol to hexanoic acid and hexanal at a pH of 4.0 [6], indicating that this enzyme has the activities of both GDH and ADH [7]. This novel GDH demonstrated great potential in processing alcoholic drinks, especially since hexanol is commonly found in wine

and liquor [8]. Having an excessive amount of hexanol in food or drink is problematic—too much hexanol not only distorts taste [9], but is also harmful to the human nervous system [10]. The Hazardous Substances Data Bank (HSDB) lists hexanol as a hazardous substance with a 50% lethal dose (LD50) of 720 mg/mL [11]. There is great potential for the use of enzymes to reduce or eliminate hexanol levels in food, with the advantages of high specificity, efficiency, and safety. Several alcohol dehydrogenases (ADHs) have the ability to oxidize hexanol, including the NADP-dependent ADHs from *Saccharomyces cerevisiae* and *Euglena gracilis* [12]. However, most of the reported ADHs are only active in neutral or alkaline conditions; this means that ADHs are typically unable to remove hexanol from alcoholic beverages, which tend to be acidic (pH of 4.5–5.5 for spirits, 2.7–3.8 for wine, and 4.0–4.4 for beer) [13]. The application of *Gc*GDH, which possesses hexanol degradation activity even in acidic conditions, would therefore be very suitable for this situation.

In our previous study, we directly isolated GDH from *G. candidum* cells, verified its identity using MALDI-TOF according to amino acid sequence, and characterized some of its basic properties. Our report was the first to describe *Gc*GDH's activity towards both higher alcohols and glutamate, indicating that the enzyme may differ from all currently reported enzymes in terms of substrate specificity. However, the intrinsic mechanism of this difference is still unclear.

Autodock cannot be used to simulate the interaction between hexanol and *Gc*GDH because of the small molecular weight and simple structure of hexanol, which has more binding sites with *Gc*GDH compared to larger molecules. Additionally, few structural studies have been performed on NADP-specific GDHs from different organisms. The current body of research has documented the crystal structures of GDH in *Escherichia coli* (strain K12) (3SBO) [14], *Clostridium symbiosum* WAL-14163 (2YFH) [15], *Escherichia coli* (strain K12) (4BHT) [16], and *Corynebacterium glutamicum* (5IJZ) [17], which are all accessible via the Protein Data Bank (PBD).

Research has also determined the crystal structures of GDH complexes with various substrates and coenzymes to assist with our understanding of the enzyme's catalytic mechanism [18,19]. The reported mammalian and bacterial GDH monomers consist of two domains: a substrate-binding domain (domain I) and an NAD/NADP-binding domain (domain II). Domain I is involved in inter-subunit interaction to form a hexamer. The substrate-binding pocket is located within the deep junction between domains I and II. Studies have also found that the binding of coenzyme or substrate induces domain closure in both bacterial and mammalian GDHs. Although several structural studies have been conducted on bacterial and mammalian GDHs, only one has focused on a fungal version of the enzyme [20]. Furthermore, NADP-GDH is poorly represented in the PDB, and the structural basis of coenzyme specificity for NADP-dependent GDHs is not well understood. The use of X-ray crystallography to examine *Gc*GDH may yield structural details on the fungal enzyme and help to improve our understanding of its functional contexts.

The purpose of this study was to uncover the intrinsic reasons that explain the special characteristics of *Gc*GDH. To accomplish this, we identified the *Gc*GDH gene, analyzed its structure, obtained the *Gc*GDH enzyme, determined its crystal structure by X-ray crystallography, located its active site, and characterized the enzyme in terms of substrate specificity and reaction conditions.

## 2. Materials and Methods

### 2.1. Microorganisms and Materials

*G. candidum* S12 (CCTCC AF2012005), which was previously isolated by our lab and kept at the China Center for Type Culture Collection (Wuhan, China), was used in this study to produce enzymes.

All chemicals were analytically pure. Methanol, ethanol, 1-propanol, n-butanol, isobutanol, n-hexanol, and isoamyl alcohol were purchased from Dr. Ehrenstorfer GmbH (Augsburg, Germany). The pET-28as vector (a pET-28a vector that fuses GDH at its N-terminus to an N-terminally polyhistidine-tagged SUMO protein tag to enhance the

solubility of the recombinant protein) was a generous gift from the College of Veterinary Medicine at Northwest A&F University. TRUE-Tag anti-His mAb Ab305-01/02 and goat anti-mouse IgG-HRP were purchased from Vazyme Biotech (Nanjing, China). The Amersham High Molecular Weight Calibration Kit for Native Electrophoresis, Superdex 200 Increase 10/300 GL, and AKTA purifier were purchased from GE Healthcare Life Sciences (Boston, MA, USA). Commercial sparse-matrix screens including Morpheus were purchased from Molecular Dimensions (Newmarket, UK). Crystal Screen, Crystal Screen Lite, Index, Natrix, JCSG core Suites I, PEG/Ion Screen, and SaltRx were purchased from Hampton Research (Aliso Viejo, CA, USA). PACT Primer HT-96 was purchased from Molecular Dimensions (Newark, NJ, USA).

### 2.2. Expression and Purification of GcGDH

The GDH sequence from *G. candidum* S12 was aligned to known sequences in the Mascot database to identify matches [7], and amplified by PCR using cDNA as a template. The primers GDH-F1 (5′-ACGCCGCTCAAGGTC-3′) and GDH-R1 (5′-TACCAA-GAAATCACCGTGGTC-3′) were used to gain the 3′ terminal sequence. Table S1 lists the 5′ race. The primers GDH F2 (5′-CGGGATCCATCAAAATGGTCCAGCCTTCC-3′) and GDH R2 (5′-CCCAAGCTTTTACCAGAAATCACCGTGGTCG-3′) were used for amplification to obtain the complete sequence of the enzyme. The PCR product was first cloned into a pMD18-T Simple Vector, restriction-digested with BamHI and HindIII to obtain the gene insert, and then ligated into BamHI/HindIII-digested pET28as. The pET28as-GDH was sequenced, transformed into *Escherichia coli* BL21 (DE3), and cultured for GDH expression [21].

*E. coli* cells containing the expression plasmid were grown overnight at 37 °C in LB with kanamycin (50 μg/mL). The resulting culture was inoculated into fresh 2 × YT at a 1:50 dilution and incubated at 37 °C until the OD at 600 nm reached 0.6. *Gc*GDH expression was induced using 0.1 mmol/L IPTG for 6 h at 25 °C.

After induction, cells were collected, washed twice, and suspended in binding buffer (following Transgen's instructions, Beijing, China), which was followed by disruption by sonication. After centrifugation and filtration through a 0.45 μm filter, the supernatant was loaded onto a His-Bind column (Ni-NTA) equilibrated with binding buffer, and the *Gc*GDH was then digested with SUMO protease to remove the SUMO and His tags. The purified *Gc*GDH was stored at 4 °C for subsequent enzymatic assays. The purified *Gc*GDH was analyzed by SDS-PAGE (12%). We performed Western blotting with the anti-6×His monoclonal antibody according to the manufacturer's instructions to identify the expressed recombinant protein.

The *Gc*GDH protein without a tag was further purified by size-exclusion chromatography using a Superdex 200 Increase 10/300 GL column with FPLC (AKTA purifier, GE healthcare, Boston, MA, USA). The peak components were collected and concentrated to ~30 mg/mL using an Amicon ultrafiltration concentrator (Massachusetts, USA) and stored at 4 °C for subsequent crystallization and enzymatic assays.

### 2.3. Crystallization

Purified *Gc*GDH protein with the His 6-tag removed (in 50 mM Tris-HCl and 100 mM NaCl at a pH of 8.0) was concentrated to approximately 30 mg/mL. The purified *Gc*GDH protein was initially crystallized using commercial sparse-matrix screens, in-cluding Morpheus, Crystal Screen, Crystal Screen Lite, Index, Natrix, PACT Primer HT-96, JCSG core Suites I, PEG/Ion Screen, and SaltRx. Crystals were grown using the sitting drop method of vapor diffusion at 20 °C for three days. Each experiment was performed with the Intelli-Plate 96-2 LVR (Art Robbins Instruments, Sunnyvale, CA, USA) by mixing 1.0 μL drops of the protein solution with 1.0 μL of the reservoir solution and equilibrating against 60 μL reservoir solution. *Gc*GDH crystals were obtained under several different conditions. We optimized the crystallization conditions to obtain the highest quality crystals possible, which appeared using Morpheus under the following conditions: 0.03 mol/L sodium

nitrate, 0.03 mol/L sodium phosphate dibasic, 0.03 mol/L ammonium sulfate, 0.1 mol/L imidazole, 0.1 mol/L MES monohydrate (acid), pH = 6.5, 20% *v/v* PEG 500*MME, and 10 % *w/v* PEG 20,000.

### 2.4. Data Collection

X-ray diffraction data were collected to a resolution of 2.3 Å at beamline BL17U1 of the Shanghai Synchrotron Radiation Facility using a MX225 CCD detector [22]. All data were indexed, integrated using auto PROC and XDS, then reduced with AIMLESS [23]. The cut-off resolution was based on the default criteria of AIMLESS. Each *Gc*GDH crystal was comprised of 12 molecules organized into an asymmetric unit, and belonged to the space group P212121 [24].

### 2.5. Structure Determination

The initial phase of the crystal structure was solved using Arcimboldo Lite through molecular replacement with combinations of polypeptide fragments and density modifications, and then model was rebuilt by ARP/wARP [25]. Maximum-likelihood-based refinement of the atomic positions and temperature factors was performed with Phenix, and the atomic model was fit with the program Coot [26]. The stereochemical quality of the final model was assessed using MolProbity [27]. Table 1 summarizes the data statistics in this experiment. The refined *Gc*GDH model was uploaded to the RCSB Protein Data Bank (PDB ID: 6IN6).

**Table 1.** Data collection and refinement statistics.

| *Gc*GDH | |
|---|---|
| Data collection | |
| Space group | P212121 |
| Unit-cell parameters (Å,°) | a = 177.611, b = 190.185, c = 202.552, α = 90°, β = 90°, γ = 90° |
| Number of reflections | 291,687 |
| Wavelength (Å) | 0.97915 |
| Resolution (Å) | 202.56-2.3059 (2.43–2.31) |
| Rmerge (%) | 21.2 (98.4) |
| I/σ(I) (last shell) | 1.75 (at 2.29 Å) |
| Completeness (%) | 97.3 (100) |
| Multiplicity (%) | 14.3 (14.5) |
| Bond lengths (Å) | 0.014 |
| Bond angles (°) | 1.509 |
| Refinement | |
| Resolution (Å) | 95.093–2.306 |
| Rwork/Rfree (%) | 27.35/32.07 |
| Bond lengths (Å) | 0.010 |
| Bond angles (°) | 1.185 |
| Ramachandran plot statistics (%) | |
| Favored/Allowed | 89.64/7.12 |
| Outliers | 3.24 |
| Error estimates | |
| Coordinate error (maximum-likelihood based) | 0.40 |
| Phase error (degrees, maximum-likelihood based) | 33.31 |

Values in Parentheses Are for the Highest Resolution Shell.

### 2.6. Determination of Enzyme Activity and Kinetic Parameters

We measured enzymatic activity towards higher alcohols according to previously described methods on the degradation of higher alcohols [28]. The AGC system (Shimadzu Scientific Instruments Inc., Kyoto, Japan) was used for measurements on the concentration of higher alcohols. Enzymatic activity towards MSG was measured at 30 °C according to the change in absorbance at 340 nm, as previously described [29]. Enzymatic activities towards methanol, ethanol, 1-propanol, isobutanol, hexanol, isoamyl alcohol, MSG, and

α-ketoglutarate were also tested to investigate the substrate specificity of *Gc*GDH. For each substrate, enzyme activity was measured using substrate concentrations of 10 mmol/L for the tested alcohols and 50 mmol/L for MSG and α-ketoglutarate.

The Michaelis constant ($K$m) and maximum velocity ($V$max) values were determined by varying the concentrations of MSG from 15 mM to 100 mM, hexanol from 10 to 50 mM, NADP$^+$ from 0.01 to 0.16 mM together with hexanol, and NADP$^+$ from 0.25 to 2.0 mM together with MSG, then constructing a Lineweaver–Burk plot.

*Gc*GDH's activity and stability were determined at various temperatures and pH values. We also examined substrate specificity and the effects of metal ions (Table 2. The catalytic reaction of *Gc*GDH was found to follow multi-substrate enzyme kinetics with a sequential mechanism of reaction.

**Table 2.** Substrate specificity of recombinant enzymes.

| Substrate | GDH Purified from pET-28as-GDH (%) Relative Activity [a] |
|---|---|
| Methanol | - |
| Ethanol | - |
| 1-propanol | - |
| 1-butanol | - |
| Isobutanol | - |
| Hexanol | 100 |
| Isoamyl alcohol | 21.39 ± 2.20 [a] |
| MSG | 100 |
| α-ketoglutarate | 140.19 ± 1.37 |

- not detected. [a] Standard deviation of triplicate measurements.

### 2.7. Mutagenesis

In order to verify the active site of *Gc*GDH, Lys 113 was replaced with Ala or Gly using a fast mutagenesis system (Transgen Biotech, Beijing, China). In Ala-F (AACATGGGTG-GTGGTGCAGGTGGTTCCG) and Ala-R (TGCACCAC-CACCCATGTTCAGGCCAGTA), the underlined alanine codon (GCA) replaced the wild-type lysine codon (AAG). In Gly-F (AACATGGGTGGTGGTGGCGGTGGTTCCG) and Gly-R (GCCACCACCACCCAT-GTTCAGGCCAGTA), the underlined glycine codon (GGC) replaced the wild-type lysine codon (AAG). After the transformed plasmid containing the mutant gene was purified and cultured in DMT competent cells, we performed single-stranded DNA sequencing to verify the successful mutation of Lys 113 to Ala or Gly. To avoid the inadvertent addition of other secondary changes, the gene was subcloned back into pET28s following mutagenesis and the entire *Gc*GDH gene was sequenced. The plasmid expressing *Gc*GDH was then overexpressed in *E. coli* BL21 (DE3). The host strain E. coli BL21 (DE3) (pET28as-Ala mutant/pET28as-Gly mutant), which contains a functional *E. coli Gc*GDH gene, was used for expression. This minimized the selective advantage for the mutant *Gc*GDH genes conferred by reversion to wild-type.

### 2.8. Reaction with Hexanol and Isoamyl Alcohol as Substrates

After the enzymatic reaction, the products converted from hexanol and isoamyl alcohol were identified using GC-MS as previously described [28].

## 3. Results and Discussion
### 3.1. Gene Cloning and Sequence Analysis

In our previous study, we identified purified *Gc*GDH from *G. candidum* S12 [7] using MALDI-TOF MS. Band matching indicated a similarity to YALIOF 17820p from *Yarrowia lipolytica* with a coverage of 10%, which was attributed to the NAD(P)-binding domain of glutamate dehydrogenase subgroup 2, while the coverage of *Gc*GDH (AHX58293.1) to homologous translation proteins of the gene sequence was about 47% and fit the mass fragment homologous to the YALIOF17820p (54%). The complete *Gc*GDH open reading

frame (ORF) was amplified using PCR, with the primers designed according to a homologous protein (YALI0F17820) gene sequence [7]. The conserved domain at the 3' end of *Gc*GDH (658 bp) was amplified separately after several failed attempts to gain the complete sequence. After two more rounds of 5'-RACE, we were able to clone the full-length sequence of *Gc*GDH. The resulting PCR product consisted of 1437 nucleotides (GenBank accession no. KJ442577.1) and was homologous to YALI0F17820p from *Yarrowia lipolytica* (XM_505553.1) in terms of both genetic sequence (77%) and amino acid sequence (78%) (Supplementary Figure S1). The longest ORF was located between nucleotides 79 and 1437, and was found to code a polypeptide 452 amino acids in length. Based on amino acid sequence, the calculated molecular mass of *Gc*GDH was 48.82 kDa. The theoretical isoelectric point of the isolated *Gc*GDH was determined to be 5.30, which is similar to the predicted value of 5.48 for the enzyme from *Yarrowia lipolytica*.

We constructed a phylogenetic tree based on the amino acid sequences of various GDHs using the neighbor-joining method [30,31]. As shown in Figure 1, the amino acid sequence of *Gc*GDH had different homologies with the reported GDH sequences of other microorganisms, as obtained from GenBank (79% to *Yarrowia lipolytica*, 79% to *Suqiyamaella liqnohabitans*, and 77% to *Kuraishia capsulate* CBS 1993). The *Gc*GDH isolated in our study was grouped in the same cluster as the protein from *Yarrowia lipolytica* (YALIOF17820; MALDI-TOF result, not listed here), belonging to the DH-like NAD(P)-binding superfamily (http://www.ncbi.nlm.nih.gov/Structure/cdd, accessed on 18 May 2017).

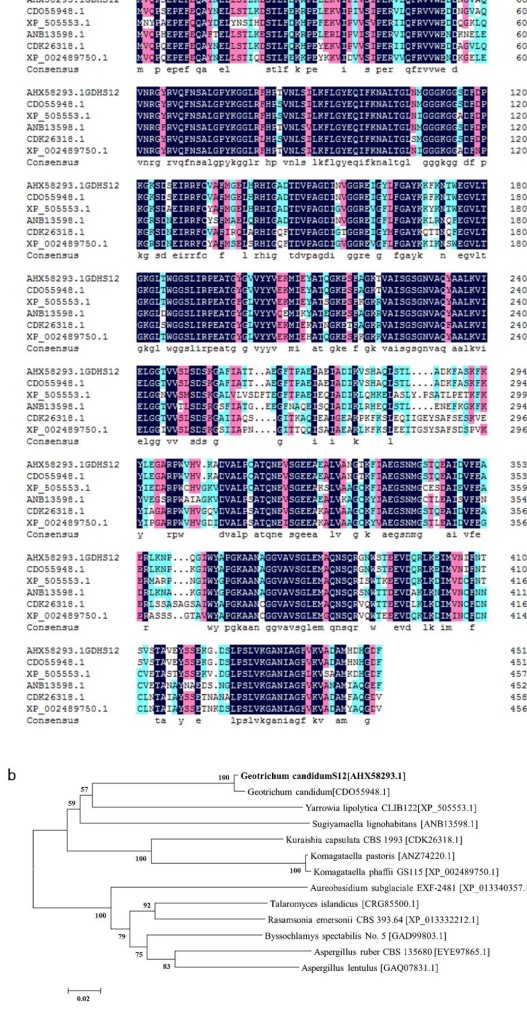

**Figure 1.** Sequence analysis. (**a**) Amino acid sequence alignment for isolated *Gc*GDH and other reported GDHs. The sequences selected for alignment were similar to those of the GDH3 NADP(+)-dependent

glutamate dehydrogenase from *Geotrichum candidum* (GenBank accession number CDO55948.1), YALI0F17820p from *Yarrowia lipolytica* CLIB122 (GenBank accession number XP_505553.1), glutamate dehydrogenase (NADP(+)) GDH1 from *Sugiyamaella lignohabitans* (GenBank accession number ANB13598.1), unnamed protein product from *Kuraishia capsulata* CBS 1993 (GenBank accession number CDK26318.1), glutamate dehydrogenase (NADP+) from *Talaromyces islandicus* (GenBank accession number CRG85500.1), and NADP(+)-dependent glutamate dehydrogenase from *Komagataellapastoris* GS115 (GenBank accession number XP_002489750.1). Identical residues are shaded in black and conserved residues are shaded in gray. (**b**) Phylogenetic tree of the glutamate dehydrogenase amino acid sequences from different organisms. Bootstrap values (%) are indicated at the nodes, and the scale bars represent 0.1 substitutions per site.

### 3.2. Expression and Purification of GcGDH

As shown in Figure 2a,b, the expressed *Gc*GDH exhibited its expected molecular mass of 50.27 kDa. Most recombinant proteins were found to be soluble. The recombinant *Gc*GDH was purified via affinity chromatography using a Ni-NTA resin column (Figure 2a). Then, the *Gc*GDH was digested with SUMO protease to remove the SUMO and His tags (Figure 2c). The *Gc*GDH protein was further purified through size-exclusion chromatography using a Superdex 200 Increase 10/300 GL column (GE Healthcare, Boston, MA, USA). The yield purified enzyme of liquid culture by the Ni-NTA resin column was found to be 27.08 mg/L. Peptide mass fingerprinting (PMF) was used to determine the purified protein fraction. Supplementary Figure S2 shows the MAIDI-TOF MS spectrum of the enzyme generated from the in-gel trypsin digestion and lists the obtained PMF peaks. The m/z values of the peptides were found to be 1121.70 (VIPIVSIPER), 1143.62 (FLGYEQIFK), 1159.61 (EIGYLFGAYK), 1222.65 (VQFNSALGPYK), 1229.64 (NPQGIWYAPGK), 1267.75 (FHPTVNLSILK), 1319.61 (GNWSTEEVDQR), 1351.60 (GNWSTEEVDQR), 1399.59 (VADAMHDHGDFW), 1599.80 (VVWEDDNGVAQVNR), 1631.79 (VVWEDDNGVAQVNR), 1648.90 (TVAISGSGNVAQYAALK), 1929.97 (AANAGGVAVSGLEMAQNSQR), 1945.96 (AANAGGVAVSGLEMAQNSQR), 2335.24 (GAFIATTAEGFTPAEIAEIADIK), 2400.13 (FIAEGSNMGSTQEAIDVFEAER), 2416.13 (FIAEGSNMGSTQEAIDVFEAER), 2452.23 (MVQPSEPEFEQAYNELLSTLK), 2468.23 (MVQPSEPEFEQAYNELL-STLK), 2614.38 (GLTWGGSLIRPEATGYGVVYYVEK), and 2646.36 (GLTWGGSLIRPEATGYGVVYYVEK). These results indicated similarity to glutamate dehydrogenase (*Geotrichum candidum*, gi | 614713480), with coverage of over 48%.

We performed several rounds of crystallization trials for *Gc*GDH using commercial screening solutions. The crystallization of proteins is known to be related to a variety of factors, such as protein concentration, ionic strength, temperature, pH, and organic solvents. Therefore, after we set the initial crystallization conditions, it was necessary for us to adjust the protein concentration, precipitant concentration, pH, temperature, size of the sitting drop, proportions of protein and reservoir in the sitting drop, and additives and protein in the buffer in order to obtain the highest quality crystals possible. After optimizing the crystallization conditions (0.03 M sodium nitrate, 0.03 M sodium phosphate dibasic, 0.03 M ammonium sulfate, 0.1 M imidazole, 0.1 M MES mono-hydrate [acid], pH = 6.5, 20% *v/v* PEG 500*MME, 10 % *w/v* PEG 20,000), we isolated the best quality protein crystal with a diffraction resolution of 2.3 Å (shown in Figure 3).

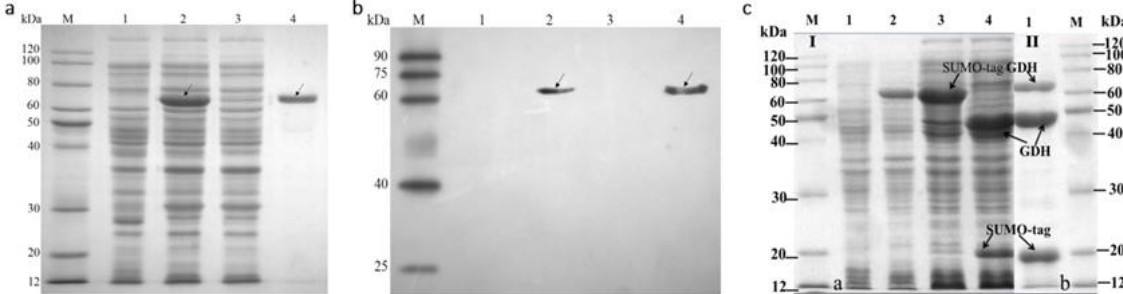

**Figure 2.** Molecular properties of purified GcGDH. (**a**) SDS-PAGE of purified GcGDH. M: protein molecular weight markers; 1: null vector transfection (control); 2: crude GcGDH; 3: penetrating crude GcGDH from Ni column; 4: purified pET-28as-GcGDH (with elution buffer containing 120 imidazole). Arrow indicates target protein. (**b**) Results of Western blot analysis. M: protein molecular weight markers; 1: null vector transfection (control); 2: crude GcGDH; 3: penetrating crude GcGDH from Ni column; 4: purified pET-28as-GcGDH (with elution buffer containing 120 imidazole). Arrow indicates target protein. (**c**) Removal of GcGDH SUMO tag by SUMO protease. aI: crude GcGDH containing the SUMO tag. M: protein molecular weight markers; 1: null vector transfection (control); a2: total expressed protein; a3: supernatant protein; a4: supernatant protein containing SUMO tag after removal by SUMO protease. bII: purified protein after passing through His-binding column. b1: purified protein containing SUMO tag cut by SUMO protease. The arrows (about 66 kDA) in Figure C a3 and Figure C b1 represent the GCGDH containing the SUMO tag; The arrows (about 50 kDA) in Figure C a4 and Figure C b1 represent the GCGDH (removal of GCGDH SUMO tag by SUMO protease); The arrows (about 16 kDA) in Figure C a4 and Figure C b1 represent the SUMO tag.3.3. Crystallization of GcGDH.

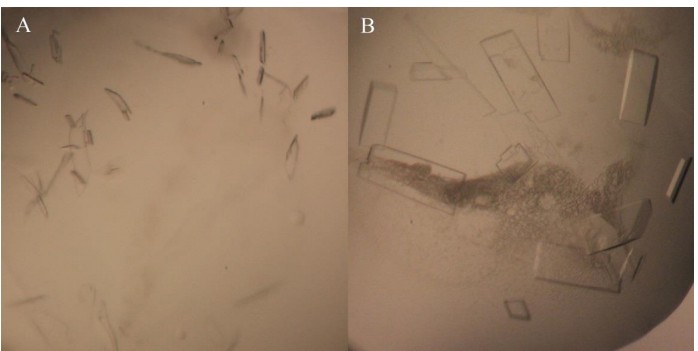

**Figure 3.** *Gc*GDH crystals were grown using the sitting drop vapor diffusion method at 20 °C. (**A**) Crystals were initially obtained, but they were small and irregular. (**B**) After optimizing the crystallization conditions, the best quality protein crystal with a diffraction resolution of 2.3 Å was isolated from the crystallization reservoir solution.

### 3.3. Data Collection and Structural Determination of GcGDH

X-ray diffraction data were collected at beamline BL17U1 of the Shanghai Synchrotron Radiation Facility using a MX225 CCD detector. The electron density map of the target protein was obtained by processing the diffraction data (Figure 4). The crystal was found to belong to space group P212121 with unit-cell parameters: a = 177.611, b = 190.185, c = 202.552, α = 90°, β = 90°, and γ = 90°. Table 1 summarizes the data collection and refinement statistics. We found that the *Gc*GDH structure consists of two hexamers (shown in Figure 5) organized into an asymmetric unit, where each subunit contains 452 amino acid residues, including 15 α-helices, 7 β-sheets, and a coil. These results were similar to those for other GDHs that contain six subunits [32]. In our study, we found that each subunit consists of an N-terminal substrate-binding domain, including α1-α7, α13, α15, and β1–β5, as well as a C-terminal cofactor-binding domain, including α8-α12, α14, β6,

and β7 (shown in Figure 5) (PDB accession code: 6IN6). These results are similar to those for other prokaryotic GDHs [15,16].

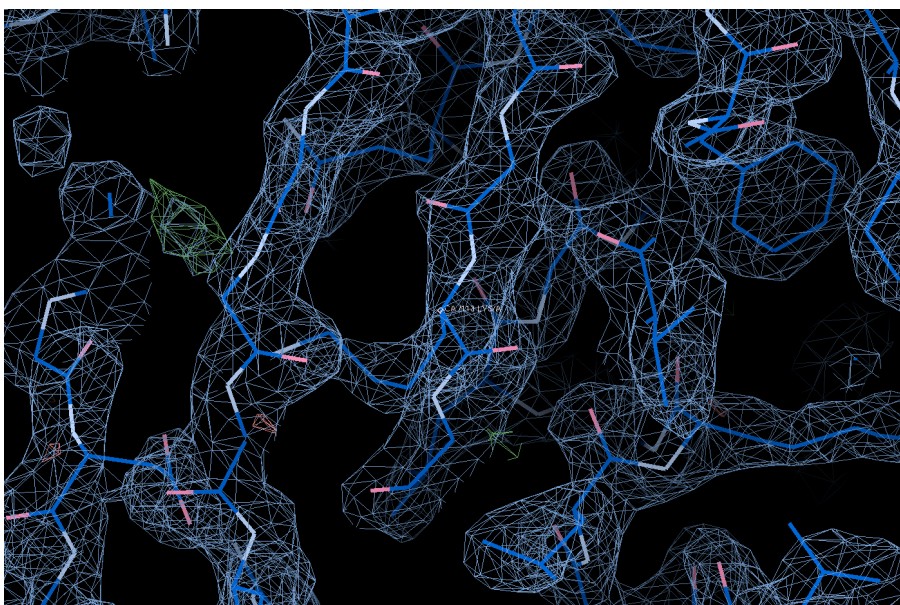

**Figure 4.** Representative portion of electron density in *Gc*GDH after refinement. The map (2*Fo–Fc*) was contoured at 1σ levels and calculated using the final model at 2.3 Å. Initially, our model was built using the amino acid sequence, corresponding parameters, and data. Then, the model was rebuilt using ARP/wARP. Atomic model fitting and refinement were performed with Phenix and Coot. The blue, white, and pale red colored bonds represent carbon, nitrogen, and oxygen atoms, respectively.

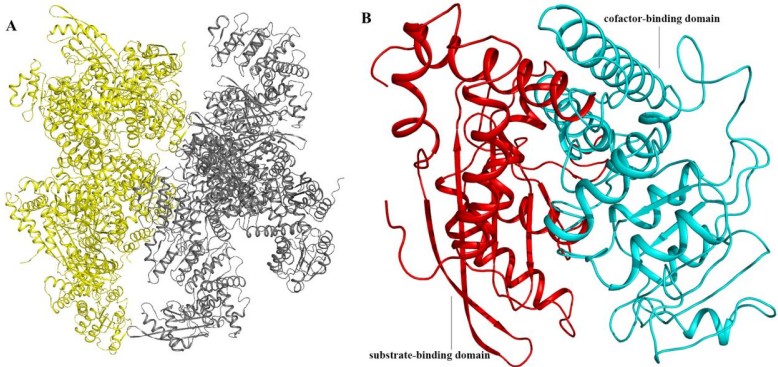

**Figure 5.** (**A**) *Gc*GDH structure comprised of 12 subunits organized into an asymmetric unit. (**B**) Each subunit consists of 452 amino acids (including 15 α-helices, 7 β-sheets, and a coil), which can be also divided into 2 domains: an N-terminal substrate-binding domain, which includes residues 1–189 and 429–452, (red; α1–α7, α13, α15, and β1-β5), and a C-terminal cofactor-binding domain, which consists of residues 190–428 (cyan; α8-α12, α14, β6, and β7).

### 3.4. Comparative Analysis of GcGDH

In order to analyze the differences between homologous *Gc*GDH proteins, we used T-coffee to compare the primary sequence of *Gc*GDH to various homologous proteins from mammals (*Homo sapiens* and *Bos Taurus*), fungi (*Aspergillus niger* and *Saccharomyces cerevisiae*), bacteria (*Corynebacterium glutamicum* and *Escherichia coli*), plants (*Chlamydomonas reinhardtii*), protists (*Plasmodium Falciparum*), archaea (*Halophilic archaeon*), and viruses (*Escherichia phage*) (Figure 6A). Our sequence analysis revealed that GDHs are conserved among different species. Still, the amino acid sequence of *Gc*GDH differed from other homologous proteins in terms of certain vital amino acids, and there was also a considerable difference in the N-terminals of these proteins. Differences were more apparent when

proteins from different species were compared. The amino acids that differed are generally regarded as essential for GDHs to perform normal biological functions such as alcohol degradation, which may explain the unique function of *Gc*GDH. Additionally, we found relatively high homology between *Gc*GDH and homologous GDHs from various microorganisms (*Aspergillus niger*, *Corynebacterium glutamicum*, *Escherichia coli*, and *Plasmodium falciparum*) (Supplementary Figure S3). The highest homology existed between *Gc*GDH and AnGDH from *Aspergillus niger*—amino acid sequence alignment indicated 73% homology (Supplementary Figure S3). We also constructed a phylogenetic tree of GDHs using Mega7 with the maximum likelihood method (Figure 6). The phylogenetic tree suggested that GDHs are present in many species, ranging from lower organisms (protozoa) to higher organisms (Chordata) (Figure 6B). Altogether, the strict evolutionary conservation of GDH across many different species suggests its importance to life.

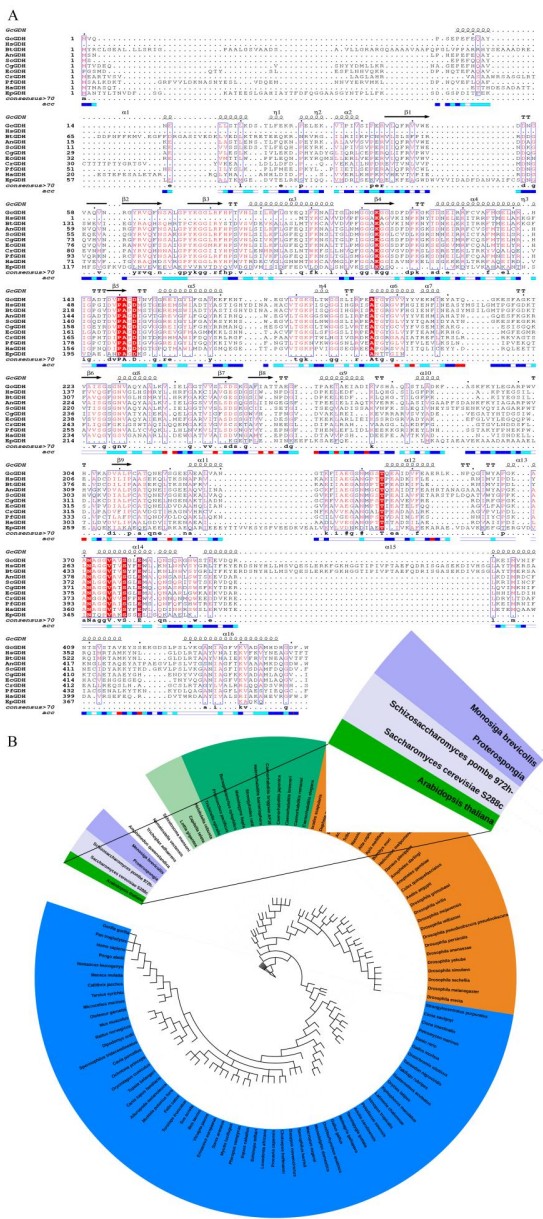

**Figure 6.** (**A**) T-coffee was used to perform multiple sequence alignment (MSA) between the target protein *Gc*GDH and its homologous proteins. The results indicated that differences between GDHs

were more notable when the proteins were from different species. The proteins are abbreviated as follows: *Gc*GDH (GDH from *Geotrichum candidum* AHX58293.1), HsGDH (GDH from *Homo sapiens* NP_001305830.1), BtGDH (GDH from *Bos taurus* NP_872593.2), AnGDH (GDH from *Aspergillus niger* 5XVI-apo), ScGDH (GDH from *Saccharomyces cerevisiae* AAA34642.1), CgGDH (GDH from *Corynebacterium glutamicum* 5IJZ_A), EcGDH (GDH from *Escherichia coli* 4FCC_A), CrGDH (GDH from *Chlamydomonas reinhardtii* XP_001694545.1), PfGDH (GDH from *Plasmodium Falciparum* 2MBA_A), HaGDH (GDH from *Halophilic archaeon* AEN06106.1), and EpGDH (GDH from *Escherichia phage* YP_007348518.1). The MSA results also revealed that there were many conserved domains among the GDHs, which are important to forming the secondary structure in many species. However, some residues in these areas do differ among the different species, and especially in highly variable regions. Red boxes represent the high sequence homology between the target protein *Gc*GDH and its homologous proteins. (**B**) A phylogenetic tree of GDHs was constructed using Mega7 with the maximum likelihood method and the bootstrap value set to 500. The final distance results were displayed using the online tool iTOL. These results indicated that GDH is an ancient enzyme with strict evolutionary conservation. The enzyme commonly exists across many species, from plants to animals and protozoa to Chordata, which reveals the importance of GDH to life. Colors in the figure are coded as follows. Blue: Chordata; orange: Arthropoda; dark green: Nematoda; pale green: Annelida; white: Platyhelminthes; purple: protozoa; light purple: Fungi; and green: Plantae. All sequence data were extracted from the NCBI database.

Given the high amino acid sequence homology between the two proteins, we performed tertiary structure comparison between *Gc*GDH and AnGDH using PyMol (Figure 7). As expected, we observed a high degree of tertiary structure similarity between *Gc*GDH and AnGDH. However, the structures were not exactly the same; for example, *Gc*GDH was found to contain a total of 15 α-helices and 7 β-sheets, whereas AnGDH consisted of 16 α-helixes and 13 β-sheets [20]. As evident from primary sequence alignment using T-coffee, cofactor and substrate binding sites were consistent between *Gc*GDH and AnGDH. Additionally, some amino acids that participate in the capture of α-ketoglutarate were the same, including Lys 77, Gln 98, Lys 102, Lys 113, Asp 153, Arg 192, and Asn 340 [20]. Lys 113 is particularly important for substrate binding to *Gc*GDH. In AnGDH, the residue forms a short hydrogen bond with a water molecule closely located to the α-carbon atom of α-ketoglutarate [20].

The transamination activity of GDH in the presence of non-natural substrates is reported to be much lower than with natural substrates [33]. In this study, we found that *Gc*GDH had a lower catalytic efficiency and higher $K$m value for hexanol compared to MSG.

### 3.5. Substrate Specificity and Metal Ion Influence

We evaluated the substrate specificity of GDH using methanol, ethanol, five other higher alcohols, MSG, and α-ketoglutarate. Hexanol, isoamyl alcohol, MSG, and α-ketoglutarate were found to be substrates of GDH, but methanol, ethanol, 1-propanol, n-butanol, and isobutanol were not (Table 2). These results indicate that the GDH in our study possesses additional activity for hexanol and isoamyl alcohol substrates. Previously, GDH enzymes have been shown to perform oxidative deamination of glutamate to α-ketoglutarate and reductive amination of α-ketoglutarate to glutamate [34]. This is the first report of a GDH with activity towards both hexanol and isoamyl alcohol.

These results differ from all currently reported ADHs and GDHs. Almost all previously described ADHs have activity towards ethanol, but only a few exhibit activities towards 1-butanol, 1-pentanol, and hexanol [13,35].

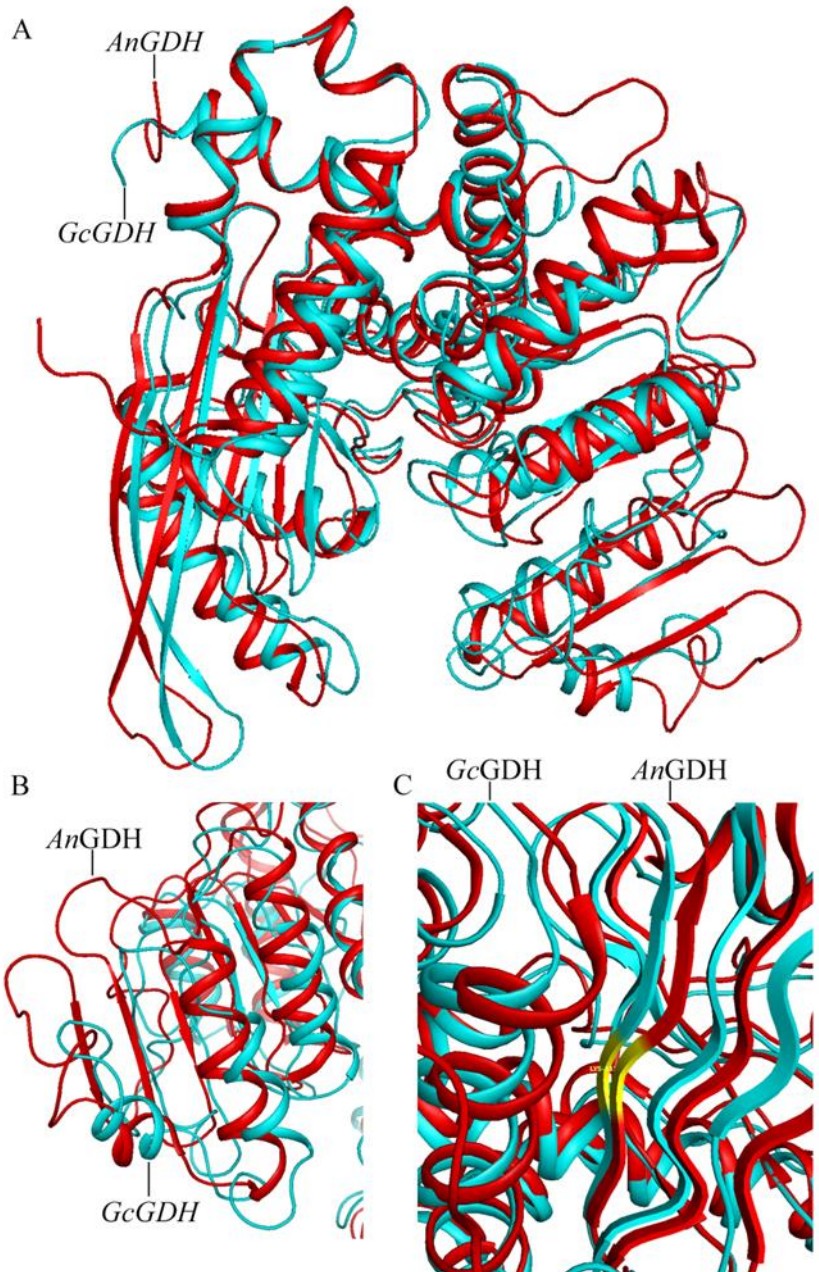

**Figure 7.** (**A**–**C**) Comparison of the tertiary structures of *Gc*GDH and AnGDH (glutamate dehydrogenase from *Aspergillus niger*) using PyMol. Although *Gc*GDH and AnGDH are highly homologous, their tertiary structures are not exactly the same. Importantly, Lys 113 (yellow) is known to be associated with substrate combination.

Next, we performed a metal-dependence assay. Without supplemental metal ions, the activity of GDH was 36.13 U/mg with glutamate and 43.72 U/mg with hexanol, and these activities were adopted as 100%. As presented in Figure 3a, enzymatic activity towards MSG was increased by $K^+$ and $Mg^{2+}$ at all tested levels in a concentration-dependent manner, with the greatest increases at 10 mmol/L (146.97%) and 20 mmol/L (210.34%) (Figure 8a). $K^+$ and $Fe^{2+}$ increased enzymatic activity towards hexanol (Figure 8b).

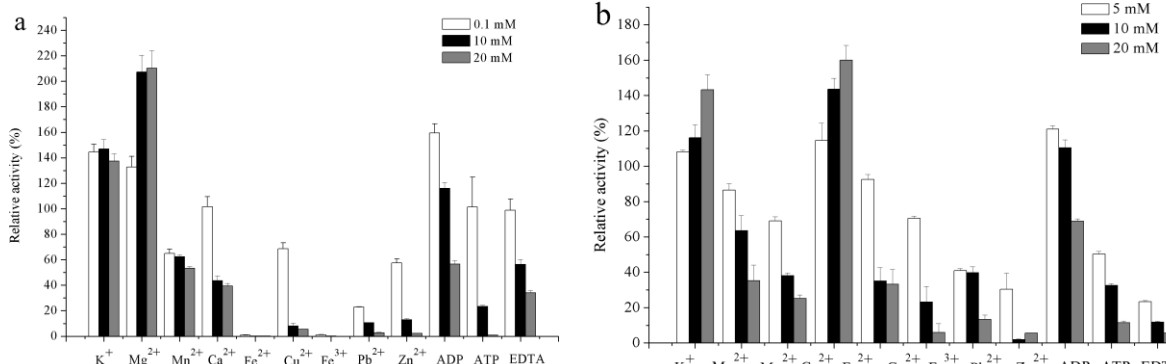

**Figure 8.** Effects of metal ions, chemical regents, and cofactors on enzymatic activity towards (**a**) MSG and (**b**) hexanol.

$Mn^{2+}$, $Fe^{2+}$, $Cu^{2+}$, $Fe^{3+}$, $Pb^{2+}$, and $Zn^{2+}$ had inhibitory effects on enzyme activity towards MSG at all tested levels (0.1–20 mmol/L; Figure 8a). $Mg^{2+}$, $Ca^{2+}$, $Cu^{2+}$, $Fe^{3+}$, $Pb^{2+}$, and $Zn^{2+}$ had the same effect on hexanol (Figure 8b). $Ca^{2+}$ had an inhibitory effect towards MSG at all tested concentrations. Other metal ions demonstrated a higher suppressing effect at higher concentrations. $Fe^{2+}$ and $Fe^{3+}$ most dramatically inhibited enzyme activity towards MSG at 20 mmol/L $Fe^{3+}$, and enzymatic activity was inhibited by 99.98% (Figure 8a). $Zn^{2+}$ had the most significant inhibitory effect on enzyme activity towards hexanol at 10 mmol/L $Zn^{2+}$, and enzymatic activity was inhibited by 98.16% (Figure 8b).

EDTA and ATP caused a decrease in enzyme activity (Figure 8a,b), except for in the case of 0.1 mmol/L ATP when hexanol was used as the substrate. When EDTA was applied, the enzymatic activities towards MSG and hexanol were only 33.95% and 5.55%, respectively. Overall, these results indicate that *Gc*GDH's enzymatic activities towards hexanol and MSG should be considered to be metal ion-dependent. The addition of ATP significantly decreased enzyme activity (Figure 8b). However, the intensity of enzyme action towards hexanol was increased in the presence of ADP, indicating that this effect may be related to dehydrogenation (Figure 8b).

*3.6. Effects of pH and Temperature on the Enzyme Activity*

GcGDH's activity towards MSG was relatively stable at pH values between 8.0 and 11, as well as at temperatures lower than 50 °C, but decreased sharply beyond these ranges (Figure 9g,h). The optimal conditions for enzyme activity were observed at a pH of 8.0 and temperature of 20 °C (Figure 9c,d). Outside of these boundaries, enzymatic activity was significantly inhibited.

*Gc*GDH's activity towards hexanol was relatively stable at a pH of 4.0 and temperatures lower than 40 °C, but sharply decreased beyond these ranges (Figure 9e,f). The optimal conditions for the enzyme's activity were observed at a pH of 4.0 or 7.0 and temperature of 30 °C (Figure 9a,b). Outside of these boundaries, enzymatic activity was significantly inhibited. Our observation that *Gc*GDH catalysis of higher alcohols is favored under acidic conditions (pH = 4.0) substantially differs from those previously reported for ADH and GDH, which favor neutral and slightly alkaline conditions [36–38]. In another study, an NAD alcohol dehydrogenase from *G. candidum* demonstrated optimized activity towards 2-propanol, 2-butanol, 2-pentanol, and 2-hexanol at a pH of 4.5–5.5, but exhibited no activity towards straight-chain alcohols [39].

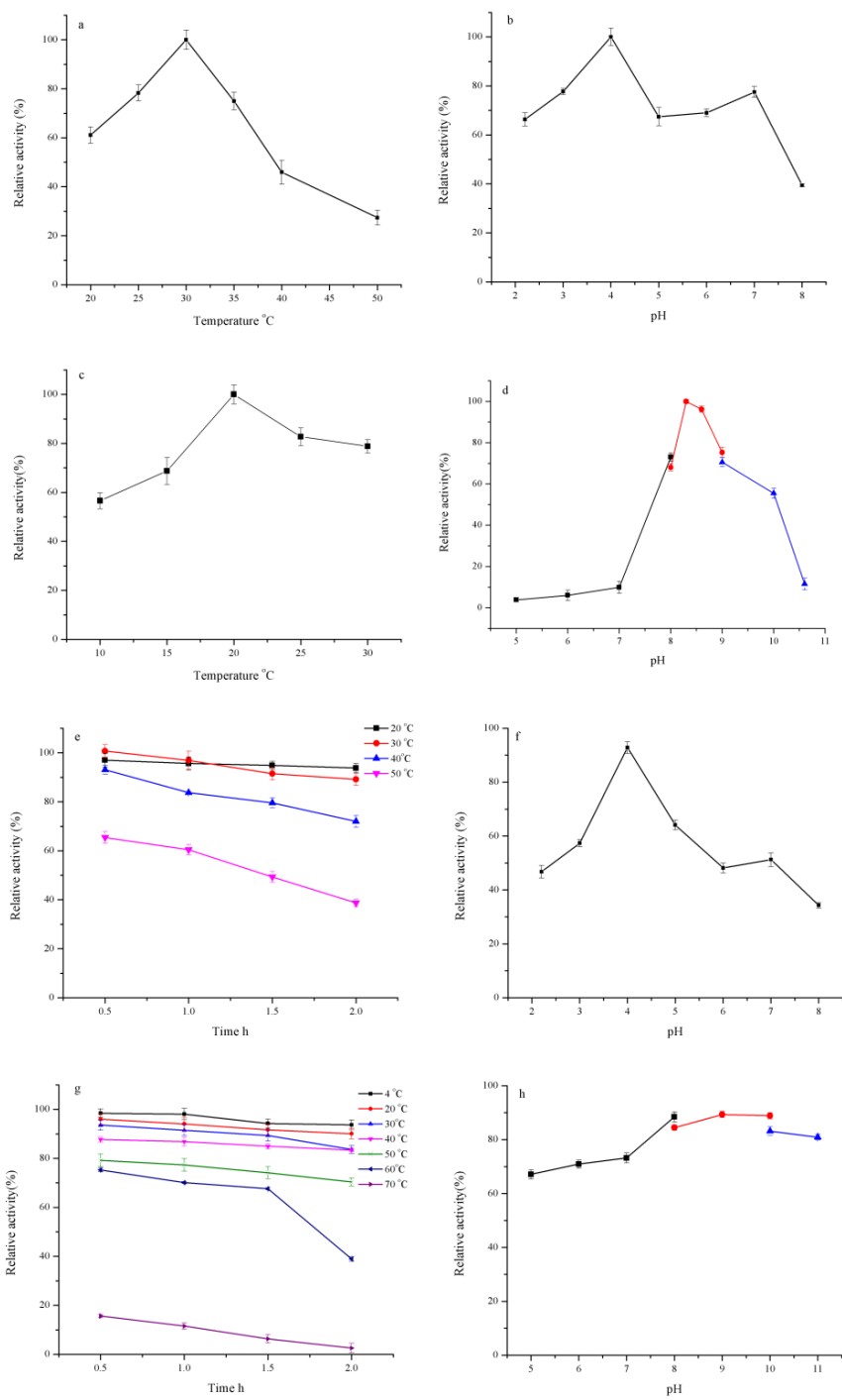

**Figure 9.** Effects of temperature and pH on the activity and stability of *Gc*GDH. (**a**) MSG and (**c**) hexanol temperature profiles. (**b**) MSG and (**d**) hexanol pH profiles. Each of the following buffers were used at 100 mM: Na$_2$HPO$_4$/citric acid (pH = 2.2–8.0), Tis-HCl (pH = 8.0–9.0), and Na2CO3/NaOH (pH = 9.0–10.5). (**e**) Glutamate and (**g**) hexanol thermal stabilities. (**f**) MSG and (**h**) hexanol pH stabilities. Each of the following buffers were used at 100 mM: Na$_2$HPO$_4$/citric acid (pH = 2.2–8.0), Tis-HCl (pH = 8–10), and Na$_2$CO$_3$/NaOH (pH = 10–11). To minimize experimental error, both the substrate solution and cuvette were heated to the corresponding temperature, and the activity measured immediately.

### 3.7. Kinetic Parameters

The kinetic parameters of *Gc*GDH's interactions with MSG and hexanol were evaluated at a pH of 8.0 and temperature of 20°C. For enzymatic activity towards MSG, the *K*m and *V*max were found to be 0.018 mmol/L and 597.65 μmol/s/mg, respectively; for enzymatic activity towards hexanol, the Km and Vmax were found to be 74.78 mmol/L and 85.47 mmol/h/mg, respectively (Figure 10a,b). In the presence of 50 mmol/L glutamate, the *K*m and *V*max of NADP were 11.11 mmol/L and 165.96 μmol/s/mg, respectively; in the presence of 10 mmol/L hexanol, these values were 42.59 mmol/L and 23.48 μmol/s/mg, respectively (Figure 10c,d).

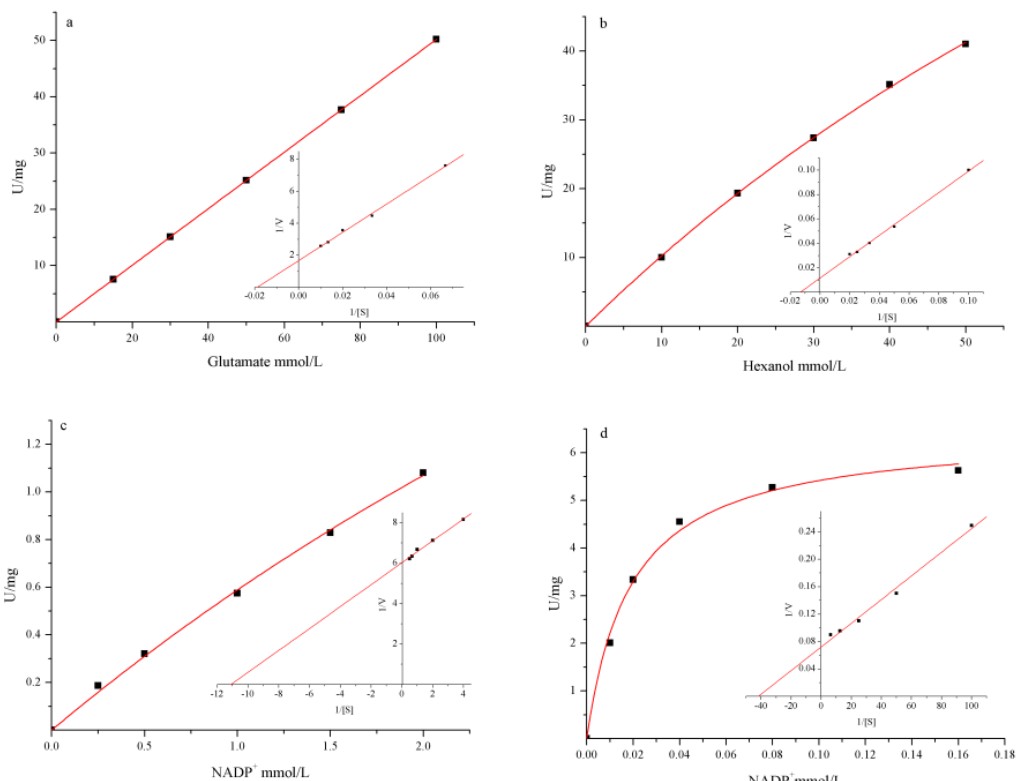

**Figure 10.** *Gc*GDH activity towards various substrates. Substrate-dependent plots for the oxidation or reduction of (**a**) MSG, (**b**) hexanol, (**c**) NADP⁺ with MSG as substrate, and (**d**) NADP⁺ with hexanol as substrate using purified *Gc*GDH (2 mg/mL). The reaction incubation time was 50 s for MSG and 1 h for hexanol. The wavelength for the substrates (MSG and NADP⁺) was 340 nm. The substrates (hexanol and NADP⁺) were analyzed using GC. Each point represents the mean of two experiments carried out in triplicates.

### 3.8. Construction of Lys 113 Mutants

In our previously study, hexanol and glutamate acted as competitive inhibitors of *Gc*GDH [7]. In order to better understand *Gc*GDH's active site, we used Scanprosite (http://prosite.expasy.org/, access on 5 June 2017) to identify the putative active site motif of the enzyme (Supplementary Figure S4a). We observed good matching to the active site of PS00074 GLFV_dehydrogenase, a Glu/Leu/Phe/Val dehydrogenase with an amino acid sequence of LNMGGGKGGSDFDP (107–120), where the underlined lysine (Lys 113) represents the active site. To test the function of the recombinant *Gc*GDH gene, we constructed two mutants using a Fast Mutagenesis System to change Lys 113 to either Ala or Gly. The variant enzymes were purified (2 mg/mL) and examined by SDS-PAGE, which revealed a single 63 kDa protein that migrated at the same position as the wild-type enzyme (Supplementary Figure S4b). We then assayed these proteins and found that the activities of the Ala 113 and Gly 113 variant proteins were 2 and 1.25 U/mg using MSG as the

substrate, respectively. These values were both significantly lower than the wild-type value of 124 U/mg (Figure 11a). Compared to the wild-type protein, we observed a signficant reduction in enzymatic activity in the Ala 113 and Gly 113 variants, with 62- and 99-fold reductions towards MSG and 175- and 3.5-fold reductions towards hexanol (Figure 11), respectively. These results suggest that Lys 113 is strictly required for catalytic activity. Furthermore, the presence of the full-length protein on the gel indicated that the absence of catalytic activity in the mutants was not due to a lack of production or stability.

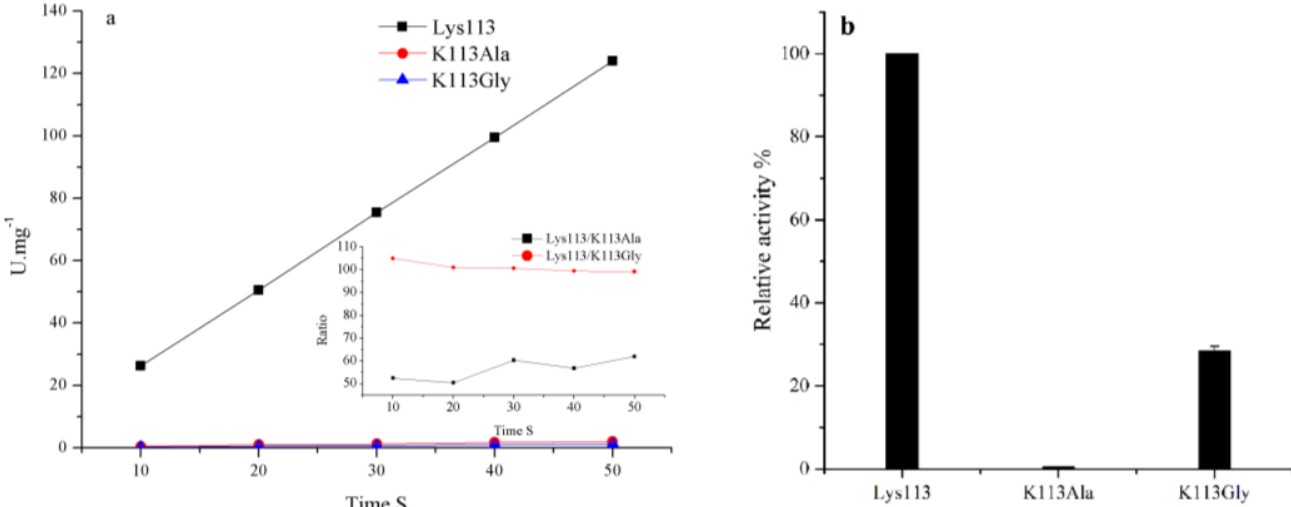

**Figure 11.** Activity of the wild-type and mutant *Gc*GDHs. (**a**) Activity of wild type *Gc*GDH and Lys 113 mutant enzymes towards MSG. (**b**) Activity of wild-type *Gc*GDH and Lys 113 mutant enzymes towards hexanol.

These results are consistent with the previous finding [17] that Lys 128 aids stabilization of the carbonyl group of the substrate via a hydrogen bond, and that this residue may also be involved in the stabilization of the amide group of glutamate when used as a substrate. In this study, the Ala 113 mutant demonstrated lower catalytic activities towards MSG and hexanol compared to the Gly 113 mutant. The activities of the Ala 113 and Gly 113 mutants were also dramatically decreased in terms of L-glutamate oxidative deamination and the degradation of higher alcohols. For this reason, we hypothesize that Lys 113 is important to both the degradation of higher alcohols and L-glutamate oxidative deamination. Perhaps the binding sites of higher alcohols with *Gc*GDH are similar to the L-glutamic acid binding site in *Gc*GDH.

Finally, we observed significant decreases in the substrate binding affinities and general GDH activities in the L113A and L113G mutants. Perhaps this is because Ala contains a methyl side group on its backbone, which could cause increased steric hindrance of the reactions. Gly is one of the residues that does not have a chiral carbon or typical flexible amino acids without net charges, resulting in low steric hindrance. Based on the previous structural and biochemical studies of GDH, the enzyme is believed to undergo an open/closed conformational change upon binding of its cofactor and substrate [17]. Future research in this field should elucidate the mechanism of *Gc*GDH activity towards higher alcohols.

### 3.9. Products of Hexanol and Isoamyl Alcohol Enzymatic Conversion

According to identification using GC-MS (Supplementary Figure S5), we found that *Gc*GDH converted hexanol to hexanal and isoamyl alcohol to 3-methyl-butanal. However, in our previous studies, hexanol was converted to hexanoic acid, hexanal, and hexyl hexanoate using a *G. candidum* S12 crude enzyme preparation [6]. This strain also exhibited the ability to convert five higher alcohols (1-propanol, n-butanol, isobutanol, hexanol, and isoamyl alcohol) to their corresponding acids and esters. These results suggest that *G. can-*

*didum* possesses additional enzymes to further process hexanal and 3-methyl butanal. In another study, furfuryl alcohol was initially oxidized into furaldehyde by NAD-ADH, which was followed by conversion to the corresponding ester by another dehydrogenase [40]. However, in P. putida, tetrahydrofurfuryl alcohol was oxidized into its corresponding carboxylic acid by NAD-ADH through two steps, in which aldehyde presented as a short-lived intermediate [41]. Geerlof studied PPQ-ADH from C. testosterone and arrived at the same conclusion, reasoning that alcohol and aldehyde competed for the same active site and the intermediate aldehyde was quickly converted to acid [42]. Here, we allowed the reaction of purified GDH to proceed at 30 °C for only 60 min. Therefore, future work should extend the reaction times to determine if additional activities would become evident for *Gc*GDH.

**Supplementary Materials:** The following supporting information can be downloaded at: https://www.mdpi.com/article/10.3390/cryst13060980/s1, Figure S1: Gene (a) and amino acid (b) sequence alignment of the GDH with *Yarrowia lipolytica*. Identical residues are highlighted in dark blue and denoted by lowercase. Highly conserved residues are denoted by colored cyan. White color depicts less conserved denoted by blank. KJ442577.1: *Geotrichum candidum* glutamate dehydrogenase (GDH) mRNA, complete cds; XM_505553.1: *Yarrowia lipolytica* YALI0F17820p (YALI0F17820g) mRNA, complete cds; AHX58293.1: GDH-S12 from G. candidum; XP_505553.1: YALI0F17820p from a *Yarrowia lipolytica* CLIB122. Figure S2: Mass spectrum obtained for tryptic peptides eluted from 1-D gel band (Figure 2a). After a baseline correction, a background subtraction, and peak deisotoping, 17 ions were submitted to Mascot. Twenty one of the submitted ions were matched to theoretical tryptic peptides from glutamate dehydrogenase; the sequences of these peptides are shown next to the mass of the monoisotopic, singly charged ions. The full protein sequence and the sequenced peptides are in red color. Figure S3: It showed that the differences of GDHs were more notable when proteins were from different microorganism. All the sequence data were extracted from NCBI database. Figure S4: The results of motif identification determined from the online tool Scanprosite (a) and SDS-PAGE of purified K113Ala (I) and K113Gly (II) (b). M: Protein molecular weight markers; 1, Crude GDH; 2, Flowed crude GDH from Ni column; 3–7, Purification processes (elution buffer containing 10-120 mM imidazole) 8, Purified pET-28as-GDH (elution buffer containing 200 mM imidazole); 5, Arrow mark indicates the target protein. Figure S5: GC-MS analysis. Total ion chromatograms of the products after GDH treatment usinghexanol and isoamyl alcohol. Table S1: List of primers used in the 5' Race. I: verify primers II:linker-adapter, III: 5'Race trans-specific primers.

**Author Contributions:** Conceptualization, J.S. and J.Z.; methodology, J.Z.; software, H.H.; validation, X.X., D.Y.; formal analysis, K.L.; writing (original draft preparation), J.Z.; visualization, C.J.; funding acquisition, D.S., J.S.; All authors have read and agreed to the published version of the manuscript.

**Funding:** This work was supported by the Modern Agricultural Industry Technology System (grant number CARS-30), the Natural Science Basic Research Plan in Shaanxi Province of China (grant number 2017JM3036), the Scientific and Technological Planning in Henan Province (grant number 212102110314), the Foundation for University Key Teacher by the Ministry of Education of Henan (grant number 2019GGJS264), the National Natural Science Foundation of Henan (grant number 212300410228), and the High-level research incubator project of College (grant numbers FCL202014), the Key Research and Development Program of Shaanxi (Program No. 2022SF-560) and the Science and Technology Innovation Team of Xinyang Agriculture and Forestry University (XNKJTD-001).

**Data Availability Statement:** The data are available from the corresponding author.

**Conflicts of Interest:** The authors declare that they have no conflict of interest.

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
