# Peer review of "Crystal Structure Analysis and Characterization of NADP-Dependent Glutamate Dehydrogenase with Alcohols Activity from Geotrichum candidum"

_crystals, doi:10.3390/cryst13060980_

Round 1
Reviewer 1 Report
Article " Crystal structure analysis and characterization of Geotrichum can-didum NADP-dependent glutamate dehydrogenase alcohol activity" (Authors: Jing Zhu and at. all) The work established the reason for the specific nature of GcGDH. Authors identified the GcGDH gene, analyzed its structure, obtained the GcGDH enzyme, determined its crystal structure by X-ray crys-88 tallography, located its active site, and characterized the enzyme in terms of substrate 89 specificity and reaction conditions. The article corresponds to the subject of the journal " Crystals", The article is well structured, written in a clear and understandable language, the conclusions are logical, the literature corresponds to the stated topic.
I recommend publishing your work.
Author Response
Thank you for your Suggestions.

Reviewer 2 Report
The article is written in a very understandable and concise way. The methods are well described. Normally, I would say that it is preferred to have a short conclusions paragraph at the end, for instance in order to give some prospective viewes on the topic of the reserach. But in this case the Authors have sufficiently described this element already in the introduction so I can agree that the conclusions paragraph can be ommited. However, I would understand if other reviewers would prefer to see it in the article.
I have only minor technical remarks:
Firstly, there are numerous places within the text when Celsius degrees are presented as ''oC'' instead of ''°C". This should be corrected.
Secondly, the resolution of the figures 9 and 10 should be improved, as it is hard to see the numbers on the axes of the graphs.
Author Response
Thank you for your Suggestions. We have corrected the representation of degrees Celsius and added a higher resolution image.
